# Pathological Traits Associated to Facebook and Twitter among French Users

**DOI:** 10.3390/ijerph17072242

**Published:** 2020-03-26

**Authors:** Élodie Verseillié, Stéphanie Laconi, Henri Chabrol

**Affiliations:** Centre d’Etudes et de Recherches en Psychopathologie et Psychologie de la Santé (CERPPS) EA7411, Université de Toulouse, 5, Allées Antonio Machado, 31058 Toulouse Cedex, France; elodie.verseillie@univ-tlse2.fr (É.V.); henri.chabrol@univ-tlse2.fr (H.C.)

**Keywords:** problematic use, addiction, Twitter, Facebook, psychopathology, personality

## Abstract

Background: With a growing number of users, social networking sites have been the subject of numerous recent studies, but little investigation has been given to their problematic use. Objectives: Our main objective was to study the relationship between psychopathological variables (i.e., personality traits, depressive and anxiety symptoms, and stress) and problematic Facebook and Twitter use. Participants and method: A sample of 1068 Internet users (Mage = 26.64; SD = 9.5) has been recruited online. Participants completed scales exploring problematic Facebook and Twitter use, and psychopathological variables. Results: Problematic Facebook and Twitter use were predicted by different pathological personality traits, regrouped in clusters in our study. Depressive and anxiety symptoms were also predictive of problematic Facebook and Twitter use but only stress explained problematic Facebook use. Gender differences have been observed. Discussion: This study highlights the relationship between depression, anxiety, stress, pathological personality traits, and problematic Facebook and Twitter use. Significant differences have been retrieved between these two uses and their relationship to psychopathology. Future research should also explore the causal relationship between social networking sites use and psychopathology and consider gender.

## 1. Introduction

Facebook and Twitter are among the most popular Social Networking Sites (SNS) worldwide [1]. All SNS facilitate social interactions, even if they do not have the same finalities. Twitter is more about sharing information and opinions [2] than about developing social interactions like Facebook [3]. Besides, Twitter restores anonymity sought initially through SNS [3] and seems more focused on what users say than on what they are or look like.

Several researches highlight the existence of addictive-like symptoms related to SNS use [4,5]. In recent years, the use of some SNS has been described as pathological, problematic, or even addictive. In this paper, the term of problematic use will be preferred because of the lack of consensual definition and designation [6]. Moreover, this term allows us to avoid the unreasonable pathologization of every SNS use. Problematic Facebook Use (PFU) has been defined as a “problematic behavior that is likely to be characterized by addictive-like symptoms and/or self-regulation difficulties leading to negative consequences in personal and social life” [5] (p. 263). Given the risks associated to SNS use, a growing interest has been put on PFU [7,8] while few studies focused on Problematic Twitter Use (PTU) [9,10,11]. Some authors [10] found patterns of addiction in PTU through withdrawal symptoms, salience, and desire to control the use. 

As for Problematic Internet Use (PIU) or Internet addiction, problematic SNS uses seem to be more frequent among adolescents and young adults. However, a prevalence of these problematic use is complicated to establish given methodological differences which are partially related to the lack of standardized definitions and assessment tools [12]. In occidental Europe, prevalence estimates of PFU are approximately ranged from 4.5% to 10% [13,14,15]. Regarding Twitter use, one French study revealed that 3.1% of participants (n = 822; Mage = 21.6; SD = 2.8; Sex Ratio = 0.59) have PTU [14].

Few studies focused on the relationship between PFU and psychological disorders. A French researcher [16] showed that among French Internet users with a PFU, 69% had anxiety symptoms and 48% had depressive symptoms. Depression has been frequently linked to PFU [16,17,18,19]. As for anxiety, mood disorders would have a causal relationship with PIU, as risk factors and consequences [20]. Besides, some studies highlighted the significant relationships between several pathological personality traits and PIU [21,22,23,24,25], yet no study focused on their relationships with SNS addictions, except for particular pathological traits. For example, borderline personality traits have been shown to be a risk factor for PFU [17], such as narcissistic personality traits, which also appeared to be related to Facebook and Twitter use [26,27].

In order to complete a lack of empirical data on PFU and PTU, particularly on French samples, our main objective was to explore the relationship between psychopathological variables (i.e., personality traits, depressive and anxiety symptoms, and stress) and problematic Facebook and Twitter use. Another aim of this research was to look at gender differences. 

## 2. Materials and Methods 

### 2.1. Participants

The sample consisted of 1068 participants aged from 18 to 64 years old (M = 26.64; SD = 9.5). There were 37% (n = 398) of men for 63% (n = 670) of women. The majority of participants were students (56%; n = 598). More detailed results are presented in Table 1. 

We spread our survey online with Survey Monkey through Facebook groups. These groups were randomly selected and covered a variety of topics (education, online games, sports, etc.) to reach the widest possible range of French users. For Twitter, the link of our survey has been shared with the most popular hashtags at different time periods. Participants were informed of the anonymity and confidentiality of their responses and gave their free and informed consent. Our study follows the ethical principles of the Declaration of Helsinki. Inclusion criteria were being at least 18 years old, understanding and speaking French, and having an account on Facebook and/or Twitter. 

### 2.2. Measures

The Bergen Facebook Addiction Scale (BFAS) [28] has been used to assess PFU. Given there was no French validation, we used a previous translated version [14]. The BFAS had six items rated on a 5-point scale ranged from 1 “Very rarely” to 5 “Very often” (e.g., How often have you used Facebook in order to forget about personal problems?). Scores varied from 5 to 30, and high scores suggest a plausible PFU. The internal consistency was α = 0.83 [28], α = 0.81 [14], and α = 0.78 in our study.

Given the lack of scale to measure PTU, we chose to use an adapted version of the BFAS [28] by replacing the term “Facebook” by “Twitter” (BFAS-T), similar to a previous study [14]. The Cronbach coefficient for the BFAS-T was α = 0.90 [14]. In this study, it was α = 0.87. 

The Depression Anxiety and Stress Scale (DASS-12) [29] has been used to assess depressive and anxiety symptoms, and stress. This scale is composed of 12 items rated on a 4-point scale from 0 “Rarely or never” to 3 “Most of the time or always”. Scores are ranged from 0 to 36. Cut off scores suggesting a severe symptomatology were respectively of 4, 4, and 6. Cronbach’ alphas were ranged between α = 0.72 and α = 0.76 [29], and between α = 0.67 and α = 0.78 in the present study. 

The Personality Diagnostic Questionnaire 4+ (PDQ4+) [30,31] measures the ten pathological personality traits of the DSM-IV with 99 items on a dichotomous scale. For this study, we used a Likert scale from 1 “Not typical to me” to 5 “very typical to me” as in previous studies [32,33]. Cronbach’ alphas for each subscale were ranged between α = 0.47 and α = 0.83 [32,33]. In our study, it varied from α = 0.38 to α = 0.94. Given this, we decided to use clusters of pathological personalities as classified in the DSM for statistical analyses. According to the DSM, Cluster A is composed of eccentric personalities (paranoid, schizoid, schizotypal), Cluster B regroups dramatic personalities (antisocial, borderline, histrionic, narcissistic), and Cluster C anxious personalities (avoidant, dependent, obsessive-compulsive). Cronbach’ alphas were α = 0.85 for Cluster A and α = 0.87 for Clusters B and C [28]. In our study, it was α = 0.78 for Cluster A, α = 0.79 for Cluster B, and α = 0.84 for Cluster C. 

### 2.3. Statistical Analyses

Descriptive analyses were conducted in the entire sample, and among Facebook and Twitter users. Pearson correlation analyses have been performed to assess the relationship between Facebook and Twitter use (including problematic use) and psychopathological variables (i.e., personality traits, depression, anxiety, stress). Multiple linear regressions helped to assess the influence of these last on Facebook and Twitter use. These analyses have also been performed separately for men and women.

## 3. Results

Among participants, 679 (64%) had a Facebook account only and 389 (36%) also had a Twitter account. No participant had only a Twitter account. As show in Table 1, the majority of participants (41%; n = 437) go to Facebook between one and five time a day and do not stay connected for long: Less than 10 min by connection (51%; n = 547). For Twitter, people connect once a day (50%; n = 193) and stay connected also less than 10 min by connection (67%; n = 262). 

Results of correlation analyses between independent variables (depression, anxiety, stress, and cluster A, B, and C of personality) are presented in Table 2 for Facebook users only and in Table 3 for Facebook and Twitter users. There is a poor but significant correlation between dependent variables PFU and PTU (r = 0.11; *p* < 0.05). 

Regression model assumptions were checked by examining the plot of predicted values of the dependent variable against residuals. It showed that assumptions were sufficiently met. Thus, among Facebook users only (n = 679), PFU has been explained by depressive symptoms (β = 0.14; *p* < 0.01), anxiety (β = 0.12; *p* < 0.05) and stress (β = 0.14; *p* < 0.01). Among Twitter users (n = 389), PTU has been explained by depressive symptoms (β = 0.17; *p* < 0.01) and anxiety (β = 0.19; *p* < 0.01). 

For personality, PFU has been explained by cluster B personality traits (β = 0.17; *p* < 0.001) and cluster C personality traits (β = 0.26; *p* < 0.001) contrary to Twitter, which has been explained by cluster A personality traits (β = 0.16; *p* < 0.05).

Correlation performed separately for men and women revealed few significant differences (see Table 4). Among women there was no significant correlation between PFU and PTU (r = 0.05; *p* = 0.50) compared to men (r = 0.22; *p* < 0.01). 

Regression analysis performed separately for men and women revealed significant differences for psychopathological variables. Among women, PFU was explained by depressive symptoms (β = 0.15; *p* < 0.05) whereas PTU was explained by anxiety (β = 0.24; *p* < 0.05). However, among men, no significant result has been retrieved for PFU and PTU, and psychopathological variables. Regarding personality, PFU was explained by cluster B and cluster C personality traits among women (β = 0.11; *p* < 0.05 and β = 0.15; *p* < 0.05) and men (β = 0.22; *p* < 0.01 and β = 0.21; *p* < 0.05), whereas PTU has been explained by cluster A personality traits (β = 0.23; *p* < 0.05) for women. PTU have not been explained by personality variables among men.

## 4. Discussion

### 4.1. Relationships with Depressive and Anxiety Symptoms

Our results are similar to those retrieved in previous studies in PFU regarding the relationship with depressive and anxiety symptoms [16,17,18,19,34,35] and PIU [36,37]. These results are not surprising and it is still necessary to fill the lack of studies on the causal relationship between these variables. The cognitive and behavioral model of problematic Internet use [20] suggests that depressive or socially anxious individuals are more prone to use Internet in order to relieve negative feelings. Young adults also experience a lot of stress in their lives due to their academic responsibilities, uncertainty about professional life or financial difficulty, which can lead to anxiety and depressive symptoms increasing the risk of PFU [34]. Using SNS in particular could induce more depressive and anxiety symptoms [38]. 

As expected, PTU was also significantly correlated to psychopathology in our study, but with a lower coefficient with stress than with PFU, among men and women. This should be explored and confirmed in other studies. 

### 4.2. Relationships with Pathological Personality Traits

Our results showed that pathological traits of Clusters B and C predicted PFU, with no significant differences among gender, as it has already been shown [34,39]. Previous works highlighted the relationship between PFU and Cluster B symptomatology such as borderline [35,40] and narcissism traits [41,42,43]. This is similar with results found with PIU for Cluster B [21,23,25,28] and Cluster C [21,25]. Borderline and antisocial traits have been frequently associated to addictive disorders, with impulsivity in common, which could be a supplementary evidence of the high similarity between PFU and other addictive disorders [44]. Besides, we can assume that borderline personalities spend a lot of time on Facebook, especially adding/searching new friends to avoid true or imagined abandonment, while narcissistic and histrionic personalities used their time on posting content hoping to receive the expected attention and compliments. Antisocial personalities might use SNS to fulfill a lack of harmonious relationships in real life with less close individuals, but also to find a satisfaction by watching others’ profiles. Cluster C traits, such as the social anxiety and the fear of being criticized or rejected in social situations, may lead to a preference for SNS with a strong control on one’s public image. The relationship between PFU and social anxiety has already been shown [45]. PFU has an effect on social anxiety and depression that decrease the level of life satisfaction [46]. Other studies showed links between PFU and obsessive-compulsive traits [38,40]. Users seems to experience craving—the urge desire to check Facebook (assimilated to compulsions), which can be explain by the fear of missing out [38]. We can suppose that dependent personality spent a lot of time on Facebook with an important Facebook involvement to maintain social relationships because of their fear of the separation and their need to be reassured. Avoidant personality traits can be easily related to PFU; avoidant individuals might feel more prone to look for online communication rather than face-to-face interaction [47]. However, similarly to what we have found, other studies found a significant relationship between PIU and Clusters B and C personalities [25,47]. Facebook appears as more similar to Internet than Twitter, which can explain some similarities.

Contrarily to PFU, PTU was predicted by Cluster A traits. This is not surprising given that Twitter restores anonymity, compared to Facebook [48]. People with Cluster A’ traits are more detached from social relationships and intimacy, and present more social anxiety and suspiciousness. Anonymity appears as initially looked for in SNS use [3] but the reduction of the social pressure and paranoid fears induced by anonymity can be the reason why Twitter use is preferred by individuals with Cluster A personality traits. Future researches should continue to study these relationships to try to confirm these results. 

### 4.3. Gender Differences

An interesting result is the lack of significant relationship between PTU and PFU among women, which highlights the specificity of these problematic SNS use as those found in gender. It could be interesting to explore if women are indeed more prone to have PFU or PTU than both as men. This result should also be interpreted given the association between other variables, such as personality traits. 

Correlation analyses revealed small differences between men and women. Among men, the correlation between Cluster B and PTU was lower than among women. PFU was also more highly related to Cluster B and C in women than men. This could suggest that problematic SNS use, in women, was related to a higher level of pathological personality traits. This is confirmed by the results of regression where no psychopathological variables explained PFU or PTU among men. Their use of SNS seems to be less related to psychopathology than for women.

Our regression analyses suggested that the relationship between PTU and cluster A is only significant among women, and that among them, PTU was explained by anxiety (and PFU by depressive symptoms). Cluster A personalities experienced more stress than other personality and it is known that women are also more at risk for anxiety symptoms [49]. Therefore, this could explain the significant relationship between PTU and cluster A, particularly among women. 

### 4.4. Implications

Even if SNS received increasing attention worldwide, the lack of data on problematic SNS use or their relationship with psychopathological variables makes this research particularly relevant. Given the increasing number of SNS users, it seems necessary to better understand SNS use and the characteristics of their users. Prevention campaign adapted for specific audience (different for Twitter and Facebook users, and probably for users of other SNS) as early detection would have a positive impact on the health of youth and could prevent people to develop a problematic use but also depressive and anxiety symptoms, as a constant pathological personality even if there is no study on the causality of these relationships. This could improve the care of these specific problematic users, with a different orientation and different therapeutic objectives according to the SNS use, its characteristics, and the presence of psychopathological symptoms and traits. 

### 4.5. Limits

Some methodological bias might have an influence on our results, such as sampling method and online recruitment. However, it seems that the use of Internet recruitment compared to the paper-and-pencil methodology has little influence on the consistency of the results [50]. Given the significant number of Facebook users, it was impossible to create a sample with only Twitter users which can make it difficult to read our results and can have an influence on them. The BFAS, which evaluates the PFU, does not have a cut off score. Thus, we have been unable to distinguish those with PFU and/or PTU from those without which did not give us the opportunity to make more precise statistics. Therefore, our assessment tools seem to be a limitation for the interpretation of our results, especially in the absence of diagnostic value. Besides, there are no consensual diagnostic criteria or definition of problematic SNS use, therefore our study lacks of a validated and diagnostic assessment tool to confirm the diagnosis of PFU and PTU such as in any study focused on this topic. Consistent in most studies, the criteria used to assess problematic SNS use are based on those of PIU even if many differences have been observed between specific and generalized Internet use [33]. This lack of specific evaluation increases the need for a unified definition and well-established and validated tools in the field of problematic SNS use. Besides, the lack of similar studies impedes our interpretation.

## 5. Conclusions

Our objectives were to explore the relationship between PFU, PTU, and psychopathological variables. This study confirmed the relationship between PFU and psychopathological variables but also brings new data about PTU. More importantly, pathological personality traits of cluster A were related to Twitter use and PTU whereas those of cluster B and C were related to Facebook use and PFU. In light of a lack of data in this area, another aim of this research was to look at gender differences in the context of PFU and PTU. Depressive and anxiety symptoms were related to PFU, whereas stress was related to PTU in both gender but with a lower intensity. Women have more anxiety symptoms and PTU than men. As a result, women seem now at a higher risk of developing a problematic SNS use. This research is one of the first to explore problematic Facebook and Twitter use and pathological personality traits. It would deserve to be replicated and deepened.

## Figures and Tables

**Table 1 ijerph-17-02242-t001:** General characteristics of the study subjects.

Samples	Total		Facebook Users		Twitter Users	
			(n = 679)		(n = 389)	
Variables	n	%	n	%	n	%
Gender						
Women	670	62.7	462	68	208	53.5
Men	398	37.3	217	32	181	46.5
Working status						
Students	598	56	383	56.4	215	55.3
Workers	397	37.2	255	37.6	142	36.5
Unemployed	73	6.8	41	6	32	8.2
Marital status						
Single	608	57	381	56	227	58.4
In relationship	460	43	298	44	162	41.6
Academic level						
School certificate	47	4.4	27	4	20	5.1
High school degree	164	15.4	92	13.5	72	18.5
Bachelor degree	496	46.4	327	48.2	169	43.4
Master	309	28.9	198	29.2	111	28.5
Doctoral degree	52	4.9	35	5.2	17	4.4
Number of visits per day						
Less than once a day	-	-	34	5	27	6.9
Between 1 and 5 times a day	-	-	289	42.6	148	38
Between 6 and 10 times a day	-	-	164	24.2	106	27.2
More than 10 times a day	-	-	192	28.3	108	27.8
Time per connection						
Less than 10 min	-	-	329	48.5	218	56
Between 10 and 30 min	-	-	236	34.8	125	32.1
Between 31 and 60 min	-	-	58	8.5	22	5.7
More than an hour	-	-	56	8.2	24	6.2
Hours per day						
One hour or less	-	-	346	51	216	55.5
Two hours	-	-	180	26.5	93	23.9
Three hours	-	-	68	10	44	11.3
Four hours	-	-	46	6.8	17	4.4
Five hours or more	-	-	39	5.6	19	5

**Table 2 ijerph-17-02242-t002:** Bivariate Pearson correlations between variables among users with an account on Facebook (n = 679).

Variables	1	2	3	4	5	6
Depression (1)	-	0.61	0.51	0.40	0.34	0.50
Anxiety (2)		-	0.59	0.36	0.32	0.47
Stress (3)			-	0.29	0.35	0.39
Cluster A (4)				-	0.59	0.60
Cluster B (5)					-	0.53
Cluster C (6)						-

Note. All correlations were significant at *p* < 0.01.

**Table 3 ijerph-17-02242-t003:** Bivariate Pearson correlations between variables among users of an account on Facebook and on Twitter (n = 389).

Variables	1	2	3	4	5	6
Depression (1)	-	0.63 **	0.56 **	0.54 **	0.41 **	0.67 **
Anxiety (2)		-	0.65 **	0.43 **	0.43 **	0.59 **
Stress (3)			-	0.43 **	0.46 **	0.51 **
Cluster A (4)				-	0.60 **	0.69 **
Cluster B (5)					-	0.57 **
Cluster C (6)						-

Note. * = *p* < 0.05; ** = *p* < 0.01.

**Table 4 ijerph-17-02242-t004:** Bivariate Pearson correlations between variables among users of an account on Facebook and on Twitter (n = 389) regarding gender.

Variables	1	2	3	4	5	6
Depression (1)	-	0.67 **	0.56 **	0.55 **	0.51 **	0.70 **
Anxiety (2)	0.58 **	-	0.64 **	0.46 **	0.48 **	0.60 **
Stress (3)	0.55 **	0.66*	-	0.39 **	0.47 **	0.48 **
Cluster A (4)	0.52 **	0.39 **	0.47 **	-	0.65 **	0.73 **
Cluster B (5)	0.32 **	0.40 **	0.47 **	0.55 **	-	0.65 **
Cluster C (6)	0.64 **	0.55 **	0.52 **	0.64 **	0.52 **	-

Note. Grey = Men; * = *p* < 0.05; ** = *p* < 0.01.

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
