# Peer review of "Pathological Traits Associated to Facebook and Twitter among French Users"

_ijerph, 2020, doi:10.3390/ijerph17072242_

Round 1

Reviewer 1 Report

The paper examines the relationship between psychopathological variables (personality traits, depressive and anxiety symptoms, and stress) and Facebook and Twitter use. This is an interesting study and I have some concerns and comments as follows.

  1. The paper lacks a description about the sample population. It will be helpful to include a table which shows the demographic distribution (age, gender, race/countries/language, education, etc.)
  2. The title of the manuscript is too big; it represents all populations or a limited sampling population?
  3. More details are needed to describe “how” you collect the data. For example, how do you send out the questions? what is the response rate?
  4. There are two different approaches in the statistical analysis, one is the correlation between two variables (Pearson’s correlation), and the other is the linear regression model. However, it is unclear which analysis is the primary analysis.
  5. In particular, you need to specify the dependent variable (PFU and PTU, I guess), and independent variables (anxiety and stress, Cluster A, B and C) if you would like to use linear regression model approach. In general, Pearson’s correlation is more commonly used (1) between dependent variables or (2) between independent variables, respectively.
  6. For relationship between dependent variable and independent variables, you better use regression model approach (both univariate regression model and multiple regression model), not Pearson’s correlation.
  7. How about the regression model assumption? For example, a normal distribution is needed for dependent variables (PFU and PTU, say).

Reviewer 2 Report

The authors introduce a topic of great interest, their goal is to identify if there are particular frailties linked to the use of specific social networks, however the study presented has numerous limits and some frailties that at present can hardly be recovered.

The main limitation lies in how the authors present the study methodology, although it is an apparently observational study, with a very low impact from the point of view of the results precisely because of the self-selection of the sample, the authors arrive at important conclusions, not supported by the described analysis.

it is not easy for the reader to understand:
- the results obtained from the use of the test on the problematic use of Facebook and twitter
- because the authors have scento these 2 social networks and not others such as Istagram
- how the population is distributed with the different personality traits in the problematic use profile

As it is known and the authors also affirm it Facebook and Twitter are 2 different types of social networks, the authors used a scale developed on Facebook, which is otherwise not valid in their country, to test the problematic use of twitter. Why should this type of ladder developed for a social network that has defined characteristics also be valid for another? validation through the use of a gold standard is required.

For these main and other minor reasons, mainly related to the clarity of the exposition, if that the authors have the opportunity to respond to all the criticisms described, the authors before submitting the work to a public health journal should completely restructure the body of their paper.

Reviewer 3 Report

In the summary should avoid mentioning the clusters. These will appear in the body of the writing, just indicate its importance.
The use of DSM IV characterizes the variables to be exercised by cluster, but does not define the disorders derived from the problem under study. Rectify these particular observations in the Materials and Methods section.
The most significant correlations respond to the weight of the predictive variable or to a coherent conceptual reduction? Clarify for tables 2 and 3.
At the close of the discussion, it is necessary to cross the starting postulates regarding the results obtained. Otherwise, the conclusions are weak.

Round 2

Reviewer 2 Report

The authors have clarified many of the doubts that I had expressed in the first review and I thank them, I believe that the paper has now improved and easier to read, however I would suggest that the authors describe the many limits of their study in a more extensive way, this in my opinion would strengthen the value of the paper which is mainly based on the originality and innovation of the work itself

Author Response

Please see the new version of the manuscript.